# Improved Artificial Neural Network Training Based on Response Surface Methodology for Membrane Flux Prediction

**DOI:** 10.3390/membranes12080726

**Published:** 2022-07-23

**Authors:** Syahira Ibrahim, Norhaliza Abdul Wahab

**Affiliations:** School of Electrical Engineering, Faculty of Engineering, Universiti Teknologi Malaysia, Johor Bahru 81310, Malaysia; syahirabintiibrahim@gmail.com

**Keywords:** feed-forward neural network, network parameters, response surface methodology, DoE, membrane biorector, palm oil mill effluent

## Abstract

This paper presents an improved artificial neural network (ANN) training using response surface methodology (RSM) optimization for membrane flux prediction. The improved ANN utilizes the design of experiment (DoE) technique to determine the neural network parameters. The technique has the advantage of training performance, with a reduced training time and number of repetitions in achieving good model prediction for the permeate flux of palm oil mill effluent. The conventional training process is performed by the trial-and-error method, which is time consuming. In this work, Levenberg–Marquardt (lm) and gradient descent with momentum (gdm) training functions are used, the feed-forward neural network (FFNN) structure is applied to predict the permeate flux, and airflow and transmembrane pressure are the input variables. The network parameters include the number of neurons, the learning rate, the momentum, the epoch, and the training functions. To realize the effectiveness of the DoE strategy, central composite design is incorporated into neural network methodology to achieve both good model accuracy and improved training performance. The simulation results show an improvement of more than 50% of training performance, with less repetition of the training process for the RSM-based FFNN (FFNN-RSM) compared with the conventional-based FFNN (FFNN-lm and FFNN-gdm). In addition, a good accuracy of the models is achieved, with a smaller generalization error.

## 1. Introduction

The palm oil mill is one of the largest industries in Asia which contributes a large amount of wastewater discharge to the waterways [1,2]. There are many advanced technologies for water and wastewater treatment nowadays, including membrane filtration technology. In the palm oil industry, the membrane bioreactor system is one of the widely used wastewater treatment technologies for palm oil mill effluent (POME). Membrane technology is preferable due to its simple operation, lesser weight, fewer space requirements, and high efficiency [3].

The submerged membrane bioreactor (SMBR) has been proven as a reliable technology in treating a wide range of water such as wastewater, groundwater, and surface water. However, fouling phenomena is the main drawback of SMBR filtration systems, which contribute to high energy consumption and maintenance costs [4]. Fouling occurs when the membrane pore is clogged by solid materials. It will cause permeate flux to be declined, and then transmembrane pressure will rise. According to [5,6,7], fouling may vary with time during the operation, and this variation can be minimized by controlling the fouling variables. Stuckey (2012) [8] showed that fouling affects a number of parameters such as the membrane form, hydrodynamic conditions, the composition of the biological system, and the operating conditions of the reactor and the chemical system.

From an operational point of view, fouling can be controlled and reduced using various techniques such as air bubble (aeration) control, backwashing, relaxation, and chemical cleaning [9]. Understanding the dynamic and prediction performance of membrane processes is very important, because with that information, the operation and control of the membrane process can be carried out more effectively in the future. Since the process of the SMBR filtration system is highly nonlinear and uncertain, it is difficult to represent it using standard mathematical equations due to the complexity of fouling behavior [8].

Artificial neural networks (ANN) have been proven reliable for the modelling of complex process involving nonlinear data. An ANN can provide a reliable and robust modelling approach, but it requires an extra step of a perturb method in the sensitivity analysis. In addition, for a chosen neural network structure, it is challenging to determine the best network parameters, which are often based on a trial-and-error basis. The proper design of ANN training (so-called ANN topology) is crucial in order to produce models with good accuracy [10,11,12]. The determination of network parameters requires a large number of its different configurations and is performed by the trial-and-error method [13,14]. The network parameters of the number of hidden layers, the neuron in the first hidden layer and the neuron in the second hidden layer [13], transfer functions, and hidden neurons [14] need to be varied until their optimal condition is determined.

Works by [15,16,17] applied a one-variable-at-time (OVAT) method, where, in this single factor optimization approach, only one factor is variable at a time, while the others are fixed by default values. This procedure is a very time-consuming and monotonous task, especially when a huge number of parameters are to be examined simultaneously [18]. The effect of the interaction between factors is also neglected, leading to a low efficiency in process optimization, which is not guaranteed to find the optimal value [19]. Works by [20,21] have successfully optimized the network setting such as the number of neurons, the epoch and training function using the genetic algorithm (GA), and differential evaluation (DE). There is no specific rule used in selecting the value of network parameters, and this process is dependent on the complexity of the modelled system.

It is most important to find the network parameters, which will affect the determination of weight and bias for model development. Several pieces of literature focus on the improvement of network training by optimizing the weights and biases using other optimization approaches such as GA and PSO [21,22]. The response surface methodology (RSM) method has successfully adopted in the literature to optimize the data from process industries, mainly for the membrane bioreactor treatment process [23,24,25]. RSM is statistical-based approach which provides a standard procedure of modelling and optimization, including the design of experiment (DoE). The DoE stage increases the accuracy of RSM model equations [26]. Furthermore, it requires less data to be collected and has the ability to detect the significant interaction effects of the factors. However, the RSM is limited to quadratic approximation and is not suitable for the approximation of high degrees of nonlinearity [27].

ANN has been proven reliable for the modelling of complex processes involving high degrees of nonlinearity [28], but it requires an extra effort in the training process. This led to works on the combination of both algorithms, which have gained much attention and have been successfully implemented as modelling and optimization tools to solve complex and nonlinear problems [26,27,28,29,30,31,32,33]. In works proposed by [27,28,32], the DoE is obtained using the RSM technique. The same DoE is utilized for both RSM (RSM-DoE) and ANN (ANN-DoE) model analyses with respect to their data accuracy and the process of analyses. From the results, the accuracy of the model obtained by ANN-DoE is preferable to that of RSM-DoE.

In other works presented by [30,34,35], the determination of ANN topology using RSM has been successfully applied, and the simulation results showed a good performance in terms of model accuracy. These ANN topologies will affect the weight and bias during ANN model development. The results showed that ANN topology, such as the step size, momentum coefficient, and training epoch, has a significant effect on the model development [34]. These studies, however, do not compare the process and the accuracy of the conventional method with the proposed (RSM) technique. So, the effectiveness of the proposed method (RSM) cannot be established. It can be observed that the concept of DoE analysis is one of the main contributions for obtaining a good accuracy of the model in an efficient way.

In this paper, an improved ANN training for feedforward neural networks (FFNN) is proposed, which utilizes the RSM-DoE-based strategy in determining the optimal conditions of network parameters. The proposed optimization is called FFNN-RSM, and the main difference, as compared to the previous work, is in terms of the ANN training parameters or topology. In this case, the training parameters are the combination of numerical and categorical factors to achieve a good model accuracy and improve the performance of training. To realize the effect of combined factors, the proposed FFNN-RSM is compared with the conventional FFNN using Levenberg–Marquardt (lm) and gradient descent with momentum (gdm) training functions. The evaluation of model performance is performed based on real input–output data from the SMBR wastewater treatment plant using palm oil mill effluent. The accuracy of the model is analyzed based on performance statistical errors such as the correlation coefficient and mean square error (MSE). The total amounts of repetition and training time are also measured and compared.

## 2. Materials and Methods

### 2.1. Experimental Methodology

Membrane bioreactor filtration treatment is carried out in a pilot scale of working volume of 30 L. The sample is collected from the final pond of the treatment plant, with a biochemical oxygen demand (BOD) of less than 10,000 mg/L, to generate fouling in the SMBR filtration process. The plant consists of a single bioreactor tank with submerged hollow fiber installed inside the tank. The hollow fiber membrane is fabricated using polyethersulfone (PES) with a pore size of approximately 80–100 kDa, and an effective membrane surface area of about 0.35 $m^{2}$ was used in the filtration system. Figure 1 shows the schematic of the pilot plant setup for the experiment, and Table 1 shows the instruments used in the pilot plant. The data plant was controlled and monitored using the National Instruments Labview 2009 software with NI USB 6009 interfacing hardware.

The total of 4000 data for each parameter were collected from the experiment, including airflow (SLPM), permeate pump (voltage), transmembrane pressure (mbar), and permeate flux (L m^−2^ h^−1^). Steps input with a random magnitude were excited for the suction pump between 0 to 3 and for the TMP between 0 to 270 mbar to obtain the dynamic behavior of the filtration process. The setting for the permeate-to-relaxation period is maintained at 120 s permeate and 30 s relaxation, with continuous aeration airflow. The airflow was set around six to eight standard liters per minute (SLPM) during filtration to maintain the high intensity of bubble flow in order to clean the membrane surface. Figure 2 shows the SMBR filtration dataset, including permeate pump voltage and TMP as the inputs and permeate flux as an output. The analysis of required data was carried out by using MATLAB R2015a and Design-Expert version 12.0.

### 2.2. Experimental Analyses

The performance of the filtration process is measured based on permeate flux as follows:(1)$$J=\frac{v}{A\times t}$$
where J is the permeate flux in ($Lm^{-2}h^{-1}$), v is the volume flow rate in liters, A is the membrane surface area ($m^{2}$), and t is the time (h).

### 2.3. Simulation Modelling

#### 2.3.1. Artificial Neural Network

The schematic structure of the FFNN is used in the present study for predicting the permeate flux of POME during membrane filtration processes, as shown in Figure 3. The input variables are transmembrane pressure and permeate pump voltage. The output variable is the permeate flux of POME. It was reported in an earlier work [36] that a network with one hidden layer and a hyperbolic tangent sigmoid (*tansig*) function is commonly used for forecasting in practice. In this work, one hidden layer with a *tansig* transfer function was considered. For the simulation process, a linear (*purelin*) function for the output was selected to produce a continuous output. The FFNN used in this study is based on the following equation:(2)$${\hat{y}}_{1}\left(t\right)=E_{i}\left[\sum_{j=1}^{nh}W_{ij}f_{j}\left(\sum_{l=1}^{n\varphi }w_{ij}+w_{j0}\right)+W_{i0}\right],$$
where ${\hat{y}}_{1}\left(t\right)$ is the prediction output. $f_{j}$ is the function of the network , and φ is the input vector. $W_{ij}$ and $w_{ij}$ represent the network connection layer weights and biases, respectively.

In this work, data normalization and data division were performed prior to the development of the neural network model. Since the input data for the SMBR system involved different magnitudes and scales, the dataset is scaled in the range of 0 to 1. This is to prevent the large original input data from dominating the solution. In addition, it prevents numerical difficulties during the calculation [37]. Equation (3) is the formula for normalization [14].
(3)$$X^{\prime }=\frac{y_{i}-y_{min}}{y_{max}-y_{min}}1,$$
where $X^{\prime }$ is the scale value, $\;y_{i}$ is the ith actual value of data, $y_{max}$ is the maximum value of data, and $y_{min}$ is the minimum value of data.

In order to investigate the feasibility of the predictive model, a combination of holdout validation and K-fold cross validation were applied. For holdout validation, all the data were randomly separated into the training dataset (*T_training_*) and testing dataset (*T_test_*). The training dataset is used for training and evaluation of the network, while the testing dataset is employed to test the performance of the network. The dataset is divided into 60% and 40% for training and testing, respectively.

K-fold cross validation is preferable for a large dataset and is chosen in this work. In this study, three-fold cross validation is used. To tune the network parameters, the training dataset is divided into a learning dataset (*T_learning_*) and validation dataset (*T_validation_*). Two thirds of the learning dataset (*T_learning_*) are the set of patterns that are used to actually train the network. The network performance is evaluated using the validation dataset (*T_validation_*), which is the remaining one-third of the pattern in the training data. The procedure was repeated three times, each time using a different fold of the observations into the learning dataset and validation dataset. Then, the average of MSECV was calculated. For the conventional method, the network parameters with a minimum validation error are selected. On the other hand, for the proposed method (FFNN-RSM), the optimal value of network parameters suggested by RSM were used. The network then finally tested on the testing dataset to yield an unbiased estimate of the performance of the network on the unseen dataset.

#### 2.3.2. Network Parameters

The network parameter is one of the main factors in achieving a good performance of a model, including FFNN model structure. Finding the best network parameters is a critical task, and appropriate ranges should be chosen. For conventional-based FFNN, the training process considers numerical factors including the number of neurons, the learning rate, the number of epochs, and the momentum coefficient training parameters. In addition to the numerical factors, the categorical factors such as training function are also considered in this paper for the better performance of model prediction.

The training functions used in this case are Levenberg–Marquardt (lm) and gradient descent with momentum (gdm). The lm training function provides great abilities such as a fast training function, a non-linear regression with a low mean square error (MSE), and memory reduction features [38,39], while the gdm has advantages such as avoiding local minima, speeding up learning, and stabilizing convergence [40]. The gdm training function depends mainly on two parameters of training: the learning rate and momentum parameters.

The former parameter aims to find the minimum weight space. Too high of a learning rate will lead to an increase in the magnitude of the oscillations for MSE, while too low of a learning rate causes smaller steps to be taken in the weight space. In this case, the capability of the network to escape from the local minima in the error surface becomes lower due to a low learning rate. The latter parameter defines the amount of momentum, where a low momentum causes less of a sensitivity of the network to the local gradient, while a high momentum causes a divergence of adaptation, which yields unusable weights. Moreover, it was found in works presented in [40,41,42] that the optimal values of the learning rate and momentum provide smooth behavior and speed up the convergence. The works also claimed that overly low values of the learning rate and momentum slowed down the converging process, while overly high values of the learning rate and momentum might lead to network instability and training divergence.

The number of neurons in the hidden layer plays a decisive role in the network performance, both for the gdm and lm training functions. For FFNN in particular, a small number of hidden neurons will cause less of an adaption in the simulation modelling. However, too big of a number of hidden neurons will cause a low ability of neural network learning, which yields system memory errors [35,40]. There is no systematic method for determining the structure (number of hidden neurons) of the network, mostly by the trial-and-error method. To achieve an accurate neural network approximation, there is an upper bound for the number of hidden neurons, as proposed by [43], which is given in Equation (4):(4)$$N^{\alpha }\leq 2N^{\beta }+1,$$
where $N^{\alpha }$ is the number of hidden neurons and $N^{\beta }\;$ is the number of inputs.

To avoid an over-fitting problem in the training data, the work in [44] proposed a relationship between the number of samples of the training data and the number of hidden neurons, given in Equation (5):(5)$$N^{\alpha }\leq \frac{N^{\theta }}{N^{\beta }+1},$$
where $N^{\theta }$ is the number of samples of the training data.

Finally, the number of epochs was selected. The number of epochs or training cycles is important to determine network models. In process modelling, a small number of epochs limits the ability of the network, while too many epochs can lead to an overtraining of the network and increase the error [35,40,45]. To train the FFNN model using different training functions (BR, LM, and GD), the maximum setting of the number of epochs is 1000 [38].

For a conventional-based FFNN, to search for optimal conditions, networks were trained under a wide range of parameter settings based on the trial-and-error method. The number of neurons in the hidden layer, the learning rate, the momentum, the number of epochs, and the training function are considered in building an optimum network structure. The optimization was carried out by applying holdout validation and K-fold cross validation on training the datasets. The training dataset was divided into two subsets in order to train the network (*T_learning_*) and to validate the network (*T_validation_*). Mean squared error cross validation (MSECV) is used to estimate the training network. The selection of corresponding FFNN parameters is based on the lowest MSECV for each parameter, which is then used for the testing dataset to verify the reliability of the network models.

The following network parameters were chosen for FFNN optimization: the number of neurons in the hidden layer, the learning rate, the momentum, the epoch number, and the training function. The range of network parameters for FFNN training, both for conventional (lm and gdm) models and the proposed (RSM) model, is shown in Table 2.

#### 2.3.3. Proposed FFNN-RSM Training Method

This section describes the proposed RSM-based FFNN (FFNN-RSM) optimization method. A complete description of the process behavior requires a quadratic or higher order polynomial model. With the use of the least square method, the quadratic models are established to describe the dynamic behavior of the process. The quadratic type of model is usually sufficient for industrial applications and, hence, is used in this work for the modelling of the permeate flux for the POME industry. For *k* factors, the following quadratic model is utilized and given in Equation (6).
(6)$$Y=\beta_{0}+\sum_{i=1}^{k}\beta_{i}x_{i}+\sum_{i=1}^{k}\beta_{ii}x_{i}x_{i}+\sum_{i=1}^{k-1}\sum_{j=i+1}^{k}\beta_{ij}x_{i}x_{j}+\varepsilon_{ij},$$
where Y is the predicted response or dependent variable, $x_{i}\;\mathrm{and}\;x_{j}$ are the independent variables, and $\beta_{i}\;\mathrm{and}\;\beta_{j}$ are the constants. The term $\beta_{0}\;$ is the intercept term, $\beta_{i}\;$ is the linear term, $\beta_{ii}\;\mathrm{and}\;\beta_{jj}$ are the squared terms, and $\beta_{ij}$ is the interaction term between the variables. The input is called a factor, and the output is known as a response.

In this case, there are five factors which consider four numerical factors (number of neurons, learning rate, momentum, and number of epoch) and one categorical factor (training function); therefore, *k* = 4 was set, which is involved for the numeric factor only. The lowest and the highest levels of variables are coded as −1 and +1, respectively, and are given in Table 2, including the axial star points of (−α and +α), where α is the distance of the axial points from the center and makes the design rotatable. In this study, the α value was fixed at 1 (face centered). The total number of experimental combinations should be conducted based on the concept of central composite design (CCD) by applying Equation (7).
(7)$$2^{k}+2k+n_{0},$$
where k is the number of independent variables (numerical factors) and $n_{0}$ is the number of experiments repeated at the center point. Then, $2^{k}$ is stated as the factorial point, while 2k is stated as an axial point. In this case, $n_{0}=6$ and k=4. Since this study involves one categoric factor with two levels (*trainlm* and *traingdm*), the number of experiments repeated at the factorial point, axial point, and center point are doubled $\left(2_{trainlm}^{4}=16,\;2_{traingdm}^{4}=16;\;2k_{trainlm}=8,\;2k_{traingdm}=8;n_{0\;trainlm}=6,\;n_{0\;traingdm}=6;\right)$. Therefore, the total number of runs needed is 60.

A matrix of 60 experiments with four numerical factors and one categorical factor was generated using the software package Design-Expert version 12.0. A total of 12 center points were used to determine the experimental error and reproducibility of the data. Table 3 shows the complete design matrix of the experiments performed and the obtained results of the MSECV. The responses were used to develop an empirical model for the permeate flux of POME. Analysis of variance (ANOVA) at a 5% level of significance, using the Fisher *F*-test, was used in determining the experimental design, interpretations, and analyses of the training data. An arithmetical method that sorts out the components of a given variation in a set of data and provides a significance test is called the *F*-test. The predicted response is transformed so that the distribution of the response variable is closer to the normal distribution. The Box-Cox plot is applied to improve model fitting. The transformations used are λ=−1, λ=0, λ=0.5, and λ=1, which respectively represent the inverse, natural log, square root, and no transformation functions [35].

The detailed methodology of the development of the proposed FFNN-RSM is depicted in Figure 4.

The proposed framework of FFNN-RSM can be described as the following steps:Divide all datasets into training datasets (*T_training_*) and testing datasets (*T_test_*);Subdivide *T_training_* into threefold: 2/3-fold data used to train the network (*T_learning_*) and 1/3-fold data used to validate the network (*T_validation_*);Create DoE settings based on RSM;Train different network parameters on *T_learning_* and evaluate its performance on *T_validation_*;Repeat step (4) for each fold and calculate the average MSECV;e.g., for no. of neuron = 1,[(MSECV_fold-1_ + MSECV_fold-2_ + MSECV_fold-3_)/3] = average MSECV_neuron-1_;Iteratively determine the network parameters by changing different network parameters based on the DoE settings;Analyze the DoE data using RSM and select the best neural network parameters;Retrain these network parameters on *T_training_*;Test for the generalization ability using *T_test_*;Compare to determine if it meets the criteria. If not, return to step (1).

### 2.4. Performance Evaluation

The performance evaluation of the model development is measured using the correlation coefficient ^®^ and mean square error (MSE), as given in Equations (8) and (9):(8)$$R=\frac{\sum_{i=1}^{n}\left(y_{i}-\bar{y}\right)\left({\hat{y}}_{i}-\bar{\hat{y}}\right)}{\sqrt{\left\{\sum_{i=1}^{n}{\left(y_{i}-\bar{y}\right)}^{2}\right\}\left\{\sum_{i=1}^{n}{\left({\hat{y}}_{i}-\bar{\hat{y}}\right)}^{2}\right\}}},$$
(9)$$MSE=\frac{1}{N}\overset{n}{\underset{i=1}{\sum }}{\left({\hat{y}}_{i}-y_{i}\right)}^{2},$$
where $y_{i}$, ${\hat{y}}_{i}$, ${\hat{y}}_{i},$ and $\bar{\hat{y}}$ denote the *i*-th independent variable, *i*-th dependent variable, the mean of dependent variables, and the mean of dependent variables, respectively. The independent and dependent variables are measured by the permeate flux and predicted permeate flux of POME, respectively. Thus, the correlation coefficient was used to assess the strength of the relationship between the inputs (permeate pump and TMP) and the permeate flux output. The MSE value near zero (0) and the R value near one (1) indicate the high accuracy of the prediction model.

## 3. Results and Discussions

This section is divided into three parts. In the first part, the results of conventional-based FFNN training in selecting neural network parameters are described. The RSM-based FFNN training (FFNN-RSM) results are presented in the second part. All data were normalized. Holdout validation and K-fold cross validation were used to ensure the robustness of the network parameters and to avoid overtraining. Finally, the model validations for permeate flux using optimal parameters from both techniques (conventional FFNN and FFNN-RSM) are presented and discussed. The accuracy of the FFNN models was measured using training and testing regressions.

### 3.1. Conventional-Based FFNN Training

This section discusses the selection of the optimum number of training parameters for the conventional FFNN. The training parameters include the number of neurons, the learning rate, the momentum coefficient, and the number of epochs. Figure 5 shows the MSECV with a varying number of neurons for FFNN-lm and FFNN-gdm. The determination of the optimum number of network parameters is based on the lowest MSECV value. The number of neurons determines the number of connections between inputs and outputs and may vary depending on specific case studies. If too many neurons are used, the FFNN becomes over-trained, causing it to memorize the training data, which affects finding the best prediction model [30,35,45,46]. In this case, the number of neurons is varied from 1 to 30, which required 30 runs. It can be seen that the optimum number of neurons was obtained at 29 and 8 for FFNN-lm and FFNN-gdm, respectively, with the lowest MSECV values of 0.0238 and 0.0533 for FFNN-lm and FFNN-gdm, respectively. The MSECV values presented by FFNN-gdm tend to fluctuate as the number of neurons varies compared with the FFNN-lm, which is more consistent, as shown in Figure 5.

Figure 6 and Figure 7 present the MSECV with varying values of learning rate and momentum, respectively. In this case, both the learning rate and momentum were trained with values set at 0.1 to 1, with 10 runs required for each parameter. As depicted in Figure 6a, it was found that the learning rates of 0.3 (MSECV = 0.0238), 0.5 (MSECV = 0.0239), and 0.7 (MSECV = 0.0232) would probably produce good results for FFNN-lm. However, the learning rates for conventional-based FFNN were obtained at 0.7 and 0.2 for lm and gdm, respectively, which gave the lowest MSECV of 0.0232 and 0.3403 for lm and gdm, respectively.

Figure 7 shows the optimum values of the momentum coefficient for FFNN-lm and FFNN-gdm. Both lm and gdm provide same value of momentum at 0.5. At this momentum value, the lowest MSECVs of 0.0243 and 0.0297 were obtained for lm and gdm, respectively. It can be observed in Figure 7b that the MSECV values of FFNN-gdm are slightly consistent at minimum values for the number of momentums at 0.1 until 0.7, with MSECVs of 0.0313, 0.0334, 0.0308, 0.0351, 0.0297, 0.0334, and 0.0303, respectively. Then, the increase in the momentum values for FFNN-gdm showed a sudden increment in MSECV to almost 1.

Figure 8 shows the MSECV with varying numbers of epochs for FFNN-lm and FFNN-gdm. In this case, the number of epochs is varied from 100 to 1000, and 10 runs are required. As illustrated in Figure 8b, it was found that the number of epochs that may produce a good outcome is 600 and 1000 with MSECVs of 0.0310 and 0.0302, respectively. Therefore, the optimum values of each epoch are obtained at 300 for lm, and 1000 for gdm was selected, which gave the lowest MSECV values of 0.0234 and 0.0302, respectively.

Table 6 summarizes the optimal values of the training parameters obtained for the conventional-based FFNN (FFNN-lm and FFNN-gdm).

### 3.2. RSM-Based FFNN Training

This section presents the optimum number of training parameters obtained using the RSM-based FFNN training. In this case, the proposed FFNN-RSM method utilized a face-central composite design (CCD) of four numerical factors (number of neurons, learning rate, momentum, and number of epochs) and one categorical factor (training function), and a two-level (*trainlm* and *traingdm*) design matrix was selected. The experimental design matrix is shown in Table 3. There are 60 sets of conditions (run) which consist of 32 factorial points, 16 axial points, and 12 center points. The different conditions of the neural network parameters were designed and trained to model the best MSE performance on the *T_validation_* dataset using the Design-Expert software.

In this work, the quadratic model was chosen to yield the correlation between the neural network effective factors (input) and the response of MSECV (output). Using the Box-Cox method, the MSECV response is transformed to a natural log (ln(MSECV)), with α equal to 1. The transformation will make the distribution of the response closer to the normal distribution and improve the fit of the model to the data [35]. The quadratic model in terms of coded factors for ln(MSECV) of the FFNN-RSM model is given in Equation (10):(10)$$\begin{array}{l} \ln\left(\mathrm{MSECV}\right)=-2.840+0.3902\;A\;+0.2335\;B\;+0.3684\;C\;+0.0196\;D\;+0.9651\;E\;+0.1013\;\mathrm{AB}\\ \;\;\;-0.0805\;\mathrm{AC}-0.0390\;\mathrm{AD}\;+0.4983\;\mathrm{AE}\;-0.0448\;\mathrm{BC}\;+0.0286\;\mathrm{BD}\;+0.2241\;\mathrm{BE}\;\\ \;\;\;+0.0636\;\mathrm{CD}\;+0.3759\;\mathrm{CE}\;+0.0218\;\mathrm{DE}\;+0.4167\;A^{2}+0.2801\;B^{2}-0.2819\;C^{2}\\ \;\;\;-0.0985\;D^{2} \end{array}$$
where the A, B, C, D, and E parameters are the code values of the number of neurons, the learning rate, the momentum, the number of epochs, and the training function, respectively, as presented in Table 3. Equation (10) is used for predictions of the response at a given level of each factor. The coded equation is useful for identifying the relative impact of the factors by comparing the factor coefficients. Negative and positive values of the coefficients represent antagonistic and synergistic effects of each model term, respectively. The positive value causes an increase in the response, while the negative value represents a decrease in the response. The values of the coefficients are relatively low, due to the low values of the MSECVs responses of the system [47].

The accuracy of the RSM model is determined using ANOVA. The ANOVA contains the set of evaluation terms such as the coefficient of determination (R^2^), adjusted R^2^, predicted R^2^, adequate precision, F-value, and *p*-value, which are used to explain the significance of the model. Table 4 illustrates the ANOVA for the response surface of the quadratic model. The statistical test factor, F-value, was used to evaluate the significance of the model at the 95% confidence level [30]. The *p*-value serves as a tool to ensure the importance of each coefficient at a specified level of significance. Generally, a *p*-value of less than 0.050 showed the most significance and contributes largely toward the response. The smaller the *p*-value, the more significant the corresponding coefficient. Other values that are greater than 0.050 are less significant.

Table 4 presents the ANOVA for the response surface of the quadratic model. The highest F-value (15.35) had a *p*-value lower than 0.0001, confirming that the model is statistically significant. The lack-of-fit test for the model showed insignificance with an F-value of 0.7680 and a *p*-value of 0.7262. This indicates that the model adequately fitted the experimental data. The value of R^2^ of 0.8794 showed a good correlation between the predicted and actual values of the responses. The value of predicted R^2^ of 0.7183 is in reasonable agreement with the adjusted R^2^ of 0.8221, also indicating the significance of the model [33]. The closer the R^2^ value is to unity, the better the model will be, as it will yield predicted values closer to the actual values. The adequate precision measures the signal-to-noise ratio, and in this analysis, (14.8721) indicates an adequate signal. A ratio greater than 4 is desirable. Thus, the model can be used to navigate the design space [30].

The *p*-values less than 0.050 (A, B, C, E, AE, BE, CE) indicate significant effects of the prediction process. The statistical analysis showed that the first order effect or linear term of training functions (E) is the most significant term in the ln(MSECV) response, followed by the number of neurons, the momentum, and the learning rate. The number of epochs depicted a less significant effect on the response, with an F-value = 0.0464 and a *p*-value = 0.8305. In addition, the *p*-values of AB, AC, AD, BC, BD, CD, DE, A^2^, B^2^, C^2^, and D^2^ are greater than 0.050 and, hence, less important in the ANN training process.

Figure 9 shows the plot of response (ln(MSECV)) and the interaction factors (number of neurons, learning rate, momentum, and number of epoch) obtained from the model graph of the Design-Expert version 12.0 software. Figure 9a shows the interaction plot of the number of neurons versus the ln(MSECV) of the training function with the other interaction factors, which are constant at their midpoints. In Figure 9a, the different shapes of curves depend on the type of training function (E). It can be observed that, with an increased number of neurons (from 1 to 30) and with *traingdm* as a training function, the ln(MSECV) increases up to 0.7. This indicates that too many hidden neurons yield more flexibility for the weight adjustment and, hence, a better learning process, particularly with noise present in the system.

Almost similar trends can be seen for an increased learning rate and momentum. However, by increasing the number of epochs (i.e., from 700 to 1000), it showed less of an effect on the ln(MSECV) response from the curves depicted in Figure 9d. These findings confirmed the statistical results obtained in Table 4—the number of neurons, the learning rate, and the momentum were significant variables for ln(MSECV), while the epoch number is not trivial. There is very little change or interaction shown by ln(MSECV) for FFNN-lm compared with ln(MSECV) for FFNN-gdm. This is because the MSECV value produced by FFNN-lm is too small, since Levenberg–Marquart (lm) has advantages such as the fastest training function and a good function fitting (non-linear regression) with a lower mean square error (MSE).

The comparisons of the actual and predicted ln(MSECV) responses based on 60 runs of various conditions of the network parameters using CCD are given in Table 5. From Table 5, the overall actual values of ln(MSECV) matched with the predicted values of ln(MSECV). This indicates that the quadratic model in Equation (10) can be established to identify the relationship between the MSECV and the network parameters.

Figure 10a,b show the perturbation plots for the lm and gdm training functions, respectively. The perturbation graph is required to see how the response changes to the changes of its factor from the reference point, with other factors held at constant reference values. In this case, the reference points default at the middle of the design space (the coded zero level of each factor). Figure 10a presents good interaction variables of the ln(MSECV) response with FFNN-lm. At the center point, factors A (number of neurons), B (learning rate), and C (momentum) produce a relatively higher effect for changes in the reference point, while only a small effect is produced by factor D (number of epoch).

Notice that the optimum values of ln(MSECV) for the network parameters (A, B, C, and D) can be found at 0.0000 (coded value), as shown in Figure 10a for the lm training function and in Figure 10b for the gdm training function. In this case, the optimum values for A, B, C, and D are, respectively, 16, 0.16, 0.75, and 850, as presented in Table 6. It can be observed that both Figure 9 and Figure 10 present similar plots of ln(MSECV), but the former plot represents the interaction graph which uses the actual value of the variables, while the latter represents the perturbation graph which uses coded values. Both graphs describe the relationship of the network parameters. As shown in Table 6, the best network parameters obtained from the conventional and proposed methods are presented for all network parameters. Therefore, the optimum values suggested by RSM were as follows: number of neurons = 16, learning rate = 0.16, momentum = 0.75, and number of epochs = 850, and *trainlm* has been selected as a training function. The best network parameters for FFNN-lm had minimum MSECVs when the number of neurons, the learning rate, the momentum, the number of epoch and the training function were 29, 0.7, 0.5, 300, and *trainlm*, respectively. Furthermore, the minimum MSECV was obtained by employing the following optimum condition for the FFNN-gdm network parameters: number of neurons = 8, learning rate = 0.2, momentum = 0.5, number of epochs = 1000, and training function = *traingdm*. These optimum values were applied for predicting the permeate flux of POME.

It may be concluded that too few hidden neurons limit the ability of the FFNN-lm to model the process well. Too many hidden neurons cause over-fitting and increase the computation time. The learning rate determines the time needed to find the minimum in the weight space. Too small of a learning rate leads to smaller steps being taken in the weight space, a slow learning process, and the network being less capable of escaping from the local minima in the error surface. Too high of a learning rate leads to an increased magnitude of the oscillations for the mean square error and a resulting slow convergence to the lower error state. Moreover, if the momentum is too small, then it will lengthen the learning process.

### 3.3. Model Validation

Figure 11a shows the training results for the permeate flux outputs for the FFNN-lm, FFNN-gdm, and FFNN-RSM models. These models are plotted based on the best ANN training parameters. From Figure 11a, it can be observed that the predicted datasets for all ANN models have similar trends to the actual or measured datasets. The permeate flux models for FFNN-lm have a slightly different shape with FFNN-gdm, but it is almost similar to FFNN-RSM. This is because FFNN-RSM also uses the *trainlm* training function as a model setting.

The dotted lines in Figure 11b–d represent the (perfect result—outputs = targets), and the solid line represents the best fit linear regression between the target and the output for FFNN-lm, FFNN-gdm, and FFNN-RSM using training data, respectively. Moreover, the FFNN model trained with *trainlm* using the conventional method (FFNN-lm) showed the highest accuracy with the correlation coefficient, with the R and MSE at 0.9888 and 0.0223, respectively, followed by FFNN-RSM (0.9881 and 0.0237) and FFNN-gdm, which were 0.9851 and 0.0296, respectively (refer to Figure 11b–d). In terms of accuracy, FFNN-lm and FFNN-RSM are comparable.

The training model was then validated using the testing dataset, and good agreement with the actual dataset was achieved, as shown in Figure 12a–d. Figure 12a shows the plot of FFNN-lm, FFNN-gdm, and FFNN-RSM permeate flux models for the SMBR filtration system during testing datasets. From Figure 12b–d, it can be seen that all the models demonstrated good prediction, with a slightly higher performance of accuracy for FFNN-lm, followed by FFNN-RSM and FFNN-gdm. The FFNN-lm model resulted in 0.9873 and 0.0253 for R and MSE, respectively. The R and MSE for FFNN-RSM are, respectively, 0.9866 and 0.0265, while the performance of the FFNN-gdm testing model was 0.9847 and 0.0303 for the R and FFNN-RSM models during the training and testing datasets.

From Table 7, it was found that all FFNN models produced comparable results for both the training and testing accuracy performance. Nevertheless, in terms of the amount of repetition and training time of the proposed method and the conventional method, the proposed method (FFNN-RSM) only required 60 runs in 233 s (00:03:53) to determine the optimal value of ANN training parameters for the FFNN model. The conventional method required 60 runs for each model (a total of 120 runs), with a total training time of 543 s for both models, which is 151 s (00:02:31.02) for FFNN-lm and 392 s (00:09:03.07) for FFNN-gdm. It is proven that the RSM technique depicted a high performance and the fastest model training technique when compared with the conventional method.

Despite the well-known advantage of the neural network in predicting larger datasets, these results show that the combined FFNN-RSM model can predict well and provide a comparable result in relation to the conventional method in this case. The FFNN-RSM shows a robust generalization ability with a small generalization error. With a smaller number of repetitions, RSM is also effective in avoiding monotonous tasks, where several different network parameters must be constructed, trained, and tested. Moreover, RSM has an ability to analyze the significant parameters and interaction effects of the parameters that affect the output response, which is the MSECV of the model. Hence, RSM is observed as satisfying the requirement for the optimization of ANN training parameters in order to obtain a good prediction model.

## 4. Conclusions

The FFNN model has been successfully developed for the permeate flux of POME during the submerged membrane bioreactor (SMBR) filtration process. The combined FFNN-RSM model has been developed to determine the network parameters, and it has been compared with conventional trial-and-error models (FFNN-lm and FFNN-gdm). The model validation results showed that good validity of the training and testing models was obtained for all models. The simulation results showed the comparable performance accuracy of the FFNN-RSM model in relation to both conventional FFNN models. The optimization of the ANN training parameters for the FFNN-RSM models shows an improved training time and number of repetitions—by about 57% and 50%, respectively—compared to the conventional FFNN model. The benefit of RSM is due to the application of the design of experiment (DoE), which requires less repetition of the training process but can provide a huge sum of information. In this work, the RSM successfully determines the best network parameters for the FFNN model. Moreover, it learned the significance and the relationship between the ANN training parameters and the MSECV.

The FFNN models (FFNN-lm, FFNN-gdm, and FFNN-RSM) demonstrated good and comparable prediction models, with a slightly higher performance of accuracy for FFNN-lm (0.9873), followed by FFNN-RSM (0.9866) and FFNN-gdm (0.9847). However, FFNN-RSM only required 60 runs in 233 s to determine the optimal value of ANN training parameters. Meanwhile, the conventional method required 60 runs for each model (a total of 120 runs), with a total training time of 543 s for both models, which is 151 s for FFNN-lm and 392 s for FFNN-gdm. The FFNN-RSM technique showed an improvement over the conventional FFNN model in terms of the number of repetitions, the training time, and the estimation capabilities. This significant improvement can be later used in control system development to improve membrane operation.

## Figures and Tables

**Figure 1 membranes-12-00726-f001:**
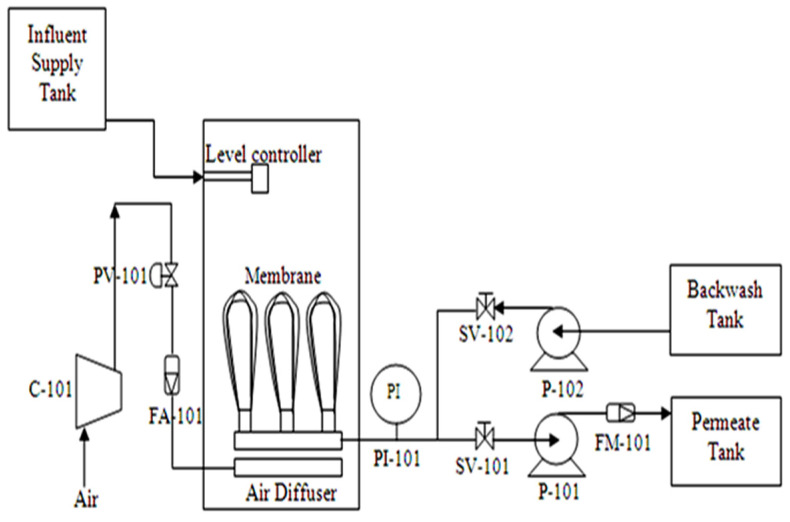
Schematic diagram of the submerged MBR (SMBR) system.

**Figure 2 membranes-12-00726-f002:**
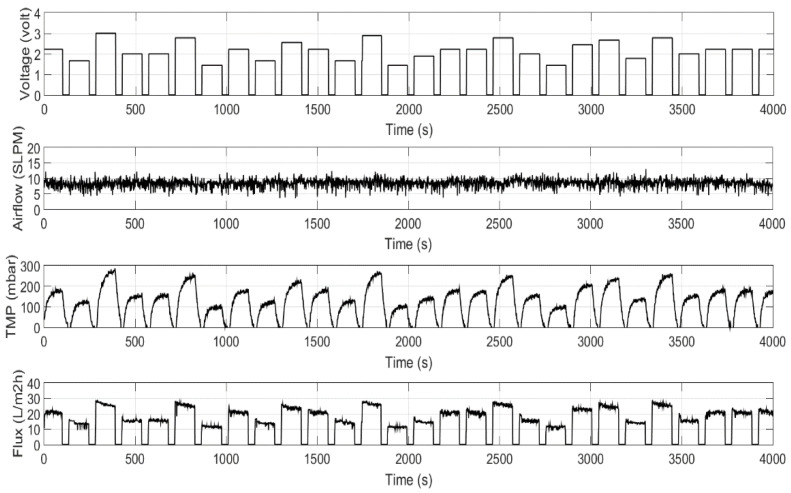
Dataset from the SMBR filtration experiment.

**Figure 3 membranes-12-00726-f003:**
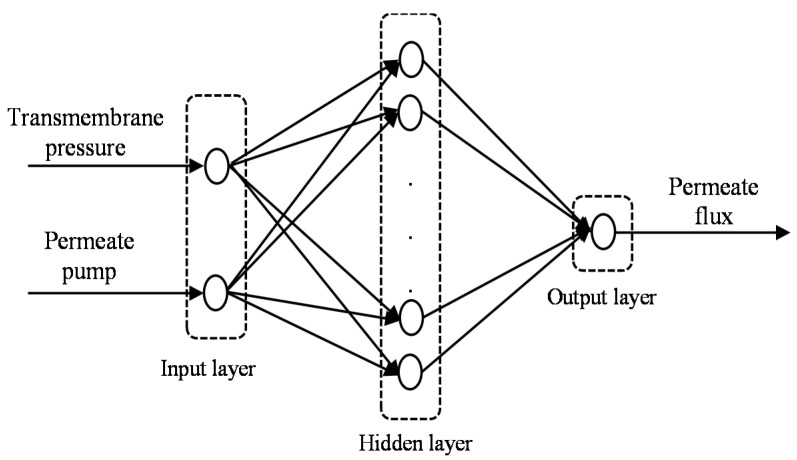
Schematic structure for FFNN.

**Figure 4 membranes-12-00726-f004:**
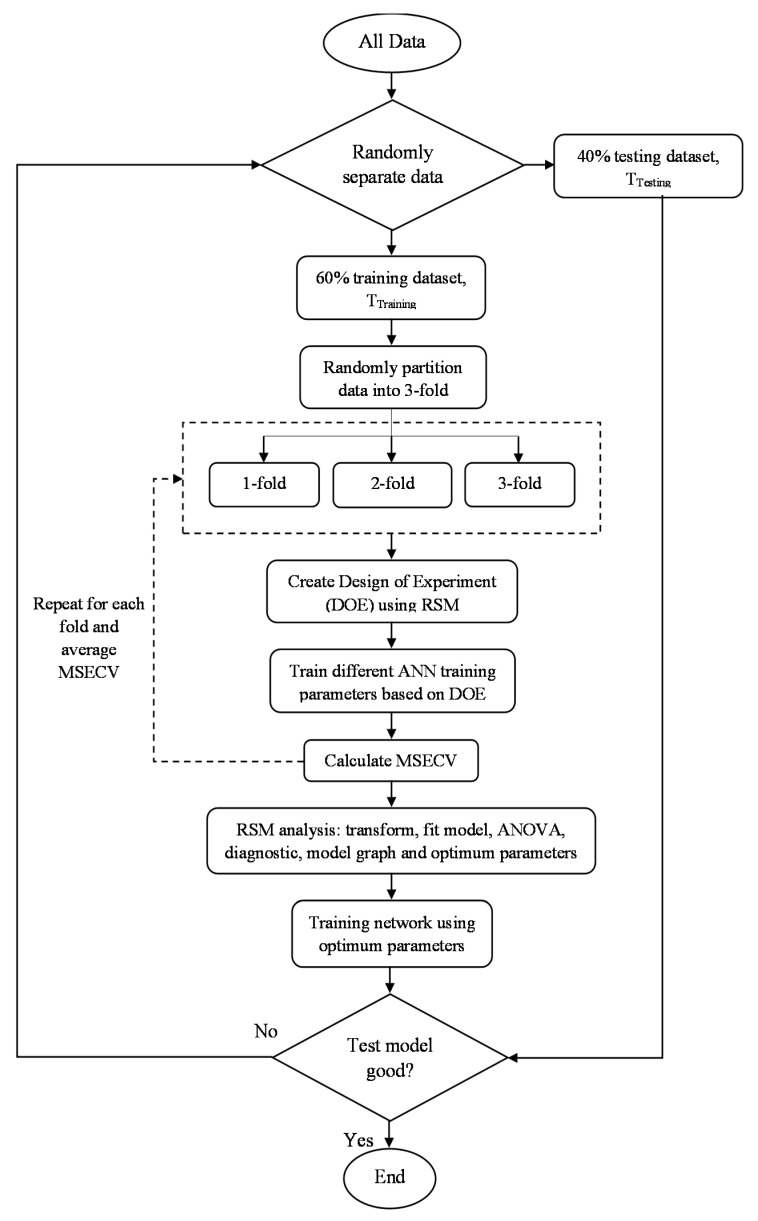
FFNN-RSM training parameters methodology.

**Figure 5 membranes-12-00726-f005:**
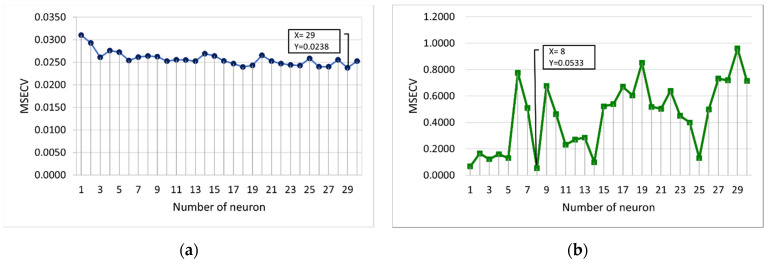
MSECV with a varying number of neurons for (**a**) FFNN-lm and (**b**) FFNN-gdm.

**Figure 6 membranes-12-00726-f006:**
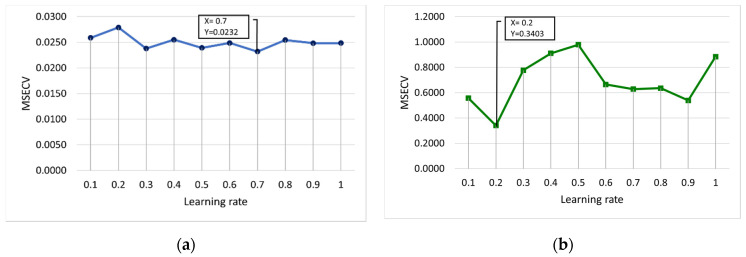
MSECV with varying values of the learning rate for (**a**) FFNN-lm and (**b**) FFNN-gdm.

**Figure 7 membranes-12-00726-f007:**
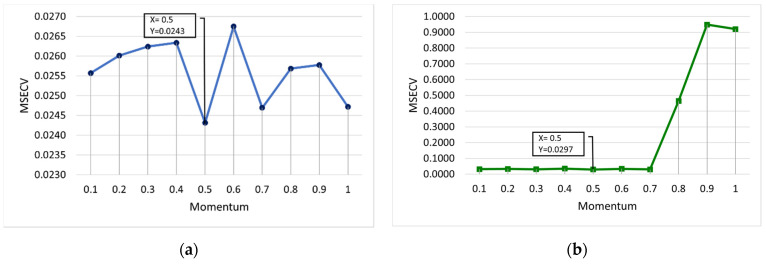
MSECV with varying values of momentum for (**a**) FFNN-lm and (**b**) FFNN-gdm.

**Figure 8 membranes-12-00726-f008:**
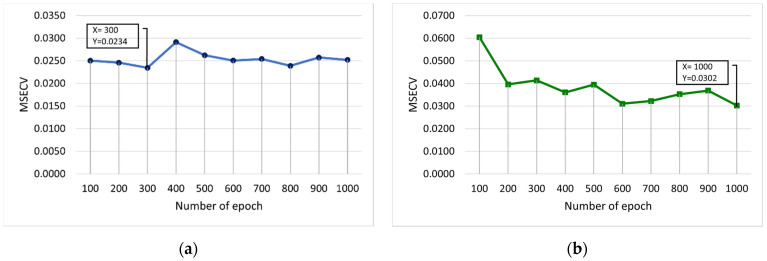
MSECV with varying numbers of epochs for (**a**) FFNN-lm and (**b**) FFNN-gdm.

**Figure 9 membranes-12-00726-f009:**
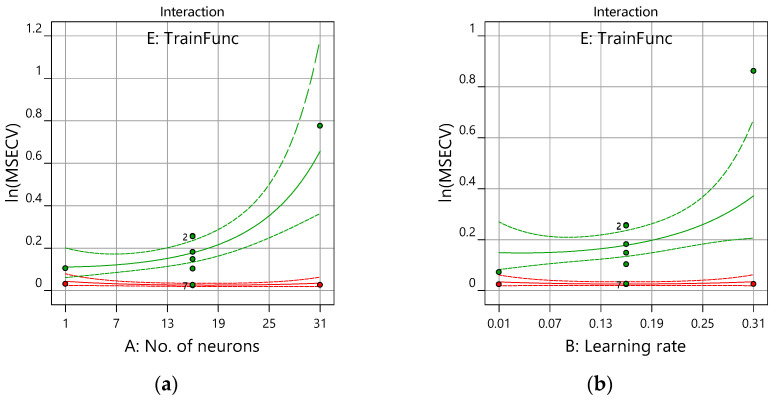
Interaction plot for the ANN training parameters of ln(MSECV) versus the (**a**) number of neurons, (**b**) learning rate, (**c**) momentum, and (**d**) number of epochs. *trainlm*—red*; traingdm*—green.

**Figure 10 membranes-12-00726-f010:**
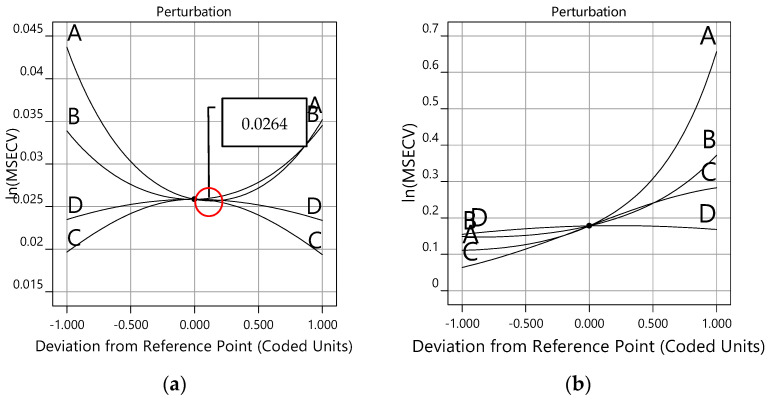
Perturbation plot for training functions, (**a**) *trainlm* and (**b**) *traingdm*.

**Figure 11 membranes-12-00726-f011:**
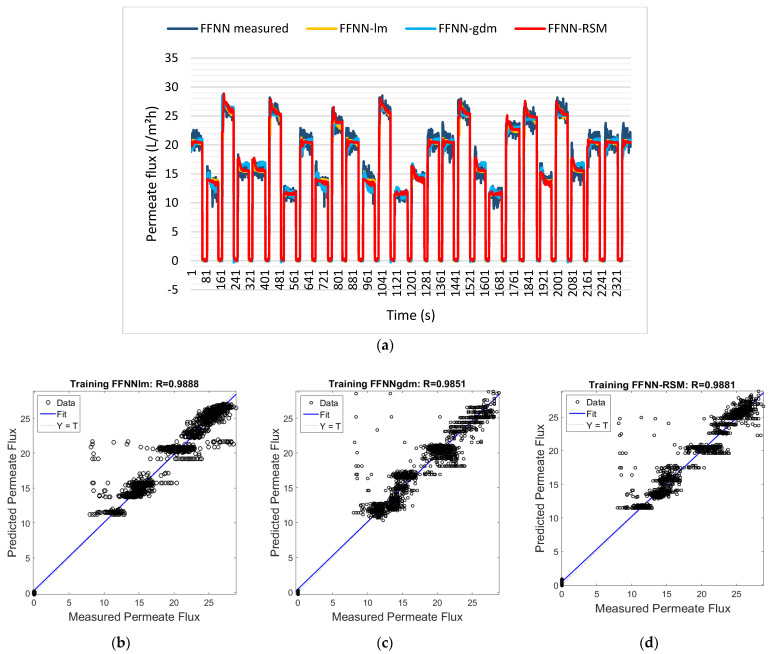
(**a**) Permeate flux for FFNN−measured, FFNN−lm, FFNN−gdm, and FFNN−RSM models. Comparison of the measured and predicted permeate flux of POME between (**b**) FFNN−lm, (**c**) FFNN−gdm, and (**d**) FFNN−RSM for the training dataset.

**Figure 12 membranes-12-00726-f012:**
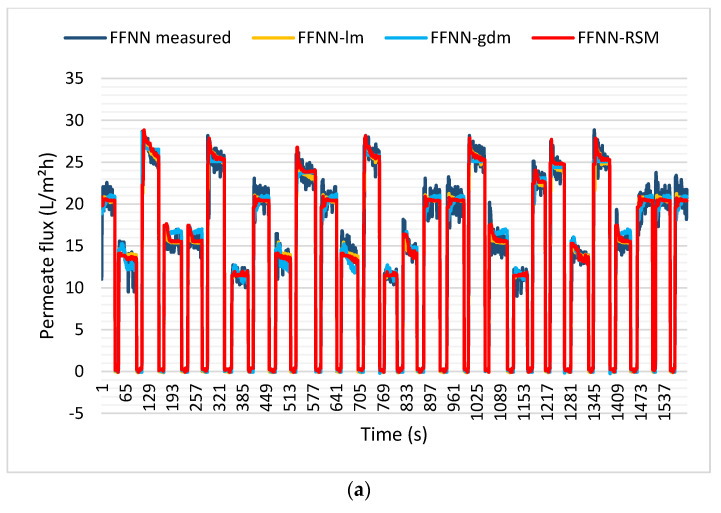
(**a**) Permeate flux for FFNN−measured, FFNN−lm, FFNN−gdm, and FFNN−RSM models. Comparison of the measured and predicted permeate flux of POME between (**b**) FFNN−lm, (**c**) FFNN−gdm, and (**d**) FFNN−RSM for the testing dataset.

**Table 1 membranes-12-00726-t001:** List of instruments in the pilot plant.

Tag No.	Description	Range
C-101	20L 2HP Air Compressor	0–20 bar
PV-101	Proportional Valve	0–5 Volt
FA-101	Airflow Sensor	0–20 SLPM
PI-101	Pressure Transducer	−1–1.5 bar
SV-101	Solenoid Valve Permeate Stream	N/A
SV-102	Solenoid Valve Backwash Stream	N/A
P-101	Peristaltic Pump	0–0.5 L min^−1^
P-102	Diaphragm Pump	0–3.8 L min^−1^
FM-101	Liquid Flow Meter	0–0.5 L min^−1^
Membrane	Hollow Fiber Membrane	80–100 kDa

**Table 2 membranes-12-00726-t002:** The range of ANN training parameters for conventional and RSM models.

Symbol	Method	Conventional		Proposed	
	Model	FFNN-lm	FFNN-gdm	FFNN-RSM	
	Network Parameters			Low (−1)	High (+1)
A	Number of neurons	1–30	1–30	1	31
B	Learning rate	0.1–1	0.1–1	0.01	0.31
C	Momentum	0.1–1	0.1–1	0.6	0.9
D	Number of epochs	100–1000	100–1000	700	1000
E	Training function	*trainlm*	*traingdm*	*trainlm*	*traingdm*
	Number of repetitions	60	60	60	

**Table 3 membranes-12-00726-t003:** MSECV results obtained by various conditions of network parameters using CCD.

Run	Space Type	A: No. of Neurons	B: Learning Rate	C: Momentum	D: No. of Epochs	E: Training Function	MSECV
1	Axial	16	0.16	0.75	1000	*trainlm*	0.0257
2	Factorial	31	0.31	0.6	1000	*traingdm*	0.9581
3	Axial	16	0.16	0.6	850	*trainlm*	0.0279
4	Axial	16	0.16	0.75	700	*traingdm*	0.1463
5	Factorial	31	0.01	0.6	1000	*traingdm*	0.1027
6	Center	16	0.16	0.75	850	*trainlm*	0.0246
7	Axial	16	0.16	0.9	850	*trainlm*	0.0250
8	Factorial	1	0.01	0.6	1000	*trainlm*	0.0310
9	Factorial	1	0.31	0.9	1000	*traingdm*	0.7342
10	Factorial	1	0.01	0.6	700	*trainlm*	0.0311
11	Center	16	0.16	0.75	850	*trainlm*	0.0243
12	Factorial	1	0.31	0.9	1000	*trainlm*	0.0309
13	Center	16	0.16	0.75	850	*trainlm*	0.0256
14	Factorial	31	0.31	0.6	1000	*trainlm*	0.0251
15	Axial	31	0.16	0.75	850	*traingdm*	0.7759
16	Center	16	0.16	0.75	850	*traingdm*	0.2568
17	Factorial	31	0.01	0.9	700	*traingdm*	0.9892
18	Factorial	31	0.01	0.9	1000	*traingdm*	0.9729
19	Factorial	1	0.31	0.6	1000	*trainlm*	0.0312
20	Center	16	0.16	0.75	850	*traingdm*	0.2548
21	Factorial	31	0.31	0.9	1000	*trainlm*	0.0238
22	Factorial	31	0.31	0.9	1000	*traingdm*	0.9859
23	Factorial	31	0.01	0.6	1000	*trainlm*	0.0250
24	Factorial	31	0.01	0.6	700	*trainlm*	0.0235
25	Factorial	31	0.31	0.6	700	*traingdm*	0.9744
26	Factorial	1	0.31	0.6	700	*traingdm*	0.0312
27	Factorial	1	0.01	0.9	700	*traingdm*	0.1039
28	Axial	16	0.01	0.75	850	*traingdm*	0.0721
29	Factorial	31	0.31	0.9	700	*traingdm*	0.9237
30	Center	16	0.16	0.75	850	*trainlm*	0.0264
31	Factorial	1	0.31	0.9	700	*trainlm*	0.0309
32	Factorial	1	0.31	0.6	1000	*traingdm*	0.0313
33	Center	16	0.16	0.75	850	*traingdm*	0.1818
34	Axial	16	0.01	0.75	850	*trainlm*	0.0243
35	Axial	16	0.31	0.75	850	*traingdm*	0.8620
36	Factorial	1	0.01	0.9	1000	*trainlm*	0.0314
37	Axial	16	0.31	0.75	850	*trainlm*	0.0256
38	Center	16	0.16	0.75	850	*trainlm*	0.0248
39	Factorial	1	0.01	0.9	700	*trainlm*	0.0309
40	Axial	31	0.16	0.75	850	*trainlm*	0.0264
41	Center	16	0.16	0.75	850	*traingdm*	0.1030
42	Factorial	31	0.31	0.9	700	*trainlm*	0.0261
43	Factorial	1	0.01	0.9	1000	*traingdm*	0.1664
44	Axial	16	0.16	0.9	850	*traingdm*	0.1913
45	Factorial	1	0.01	0.6	700	*traingdm*	0.0615
46	Axial	16	0.16	0.75	700	*trainlm*	0.0262
47	Axial	1	0.16	0.75	850	*traingdm*	0.1048
48	Axial	16	0.16	0.75	1000	*traingdm*	0.0863
49	Factorial	31	0.01	0.9	1000	*trainlm*	0.0252
50	Axial	1	0.16	0.75	850	*trainlm*	0.0311
51	Factorial	31	0.01	0.6	700	*traingdm*	0.1022
52	Factorial	31	0.31	0.6	700	*trainlm*	0.0261
53	Factorial	1	0.31	0.9	700	*traingdm*	0.2292
54	Center	16	0.16	0.75	850	*traingdm*	0.1480
55	Factorial	1	0.01	0.6	1000	*traingdm*	0.0412
56	Factorial	31	0.01	0.9	700	*trainlm*	0.0243
57	Factorial	1	0.31	0.6	700	*trainlm*	0.0312
58	Center	16	0.16	0.75	850	*trainlm*	0.0259
59	Center	16	0.16	0.75	850	*traingdm*	0.0266
60	Axial	16	0.16	0.6	850	*traingdm*	0.0306

**Table 4 membranes-12-00726-t004:** Analysis of variance (ANOVA) for the response surface of the quadratic model.

Source	Sum of Squares	df	Mean Square	F-Value	*p*-Value	
Model	87.29	19	4.59	15.35	<0.0001	significant
A: No. of neurons	5.48	1	5.48	18.32	0.0001	significant
B: Learning rate	1.96	1	1.96	6.56	0.0143	significant
C: Momentum	4.89	1	4.89	16.33	0.0002	significant
D: No. of epoch	0.0139	1	0.0139	0.0464	0.8305	
E: TrainFunc	55.88	1	55.88	186.76	<0.0001	significant
AB	0.3281	1	0.3281	1.10	0.3013	
AC	0.2072	1	0.2072	0.6924	0.4103	
AD	0.0486	1	0.0486	0.1624	0.6891	
AE	8.94	1	8.94	29.88	<0.0001	significant
BC	0.0643	1	0.0643	0.2149	0.6454	
BD	0.0261	1	0.0261	0.0874	0.7691	
BE	1.81	1	1.81	6.04	0.0184	significant
CD	0.1292	1	0.1292	0.4319	0.5148	
CE	5.09	1	5.09	17.00	0.0002	significant
DE	0.0172	1	0.0172	0.0574	0.8120	
A²	0.8998	1	0.8998	3.01	0.0906	
B²	0.4066	1	0.4066	1.36	0.2506	
C²	0.4117	1	0.4117	1.38	0.2477	
D²	0.0503	1	0.0503	0.1681	0.6840	
**Residual**	11.97	40	0.2992			
Lack of Fit	8.35	30	0.2782	0.7680	0.7262	not significant
Pure Error	3.62	10	0.3622	15.35	<0.0001	
**Cor Total**	99.26	59	4.59	18.32	0.0001	
**R^2^**	0.8794					
**Adjusted R^2^**	0.8221					
**Predicted R^2^**	0.7183					
**Adeq. Precision**	14.8721					

**Table 5 membranes-12-00726-t005:** Comparisons of the actual and predicted ln(MSECV) by various conditions of network parameters.

Run	Space Type	A: No. of Neurons	B: Learning Rate	C: Momentum	D: No. of Epochs	E: Training Function	MSECV	Actual ln(MSECV)	Predicted ln(MSECV)
1	Axial	16	0.16	0.75	1000	*trainlm*	0.0257	−3.6600	−3.9100
2	Factorial	31	0.31	0.6	1000	*traingdm*	0.9581	−0.0428	−0.7624
3	Axial	16	0.16	0.6	850	*trainlm*	0.0279	−3.5800	−4.0800
4	Axial	16	0.16	0.75	700	*traingdm*	0.1463	−1.9200	−2.0100
5	Factorial	31	0.01	0.6	1000	*traingdm*	0.1027	−2.2800	−2.0300
6	Center	16	0.16	0.75	850	*trainlm*	0.0246	−3.7100	−3.8000
7	Axial	16	0.16	0.9	850	*trainlm*	0.0250	−3.6900	−4.0900
8	Factorial	1	0.01	0.6	1000	*trainlm*	0.0310	−3.4700	−3.4600
9	Factorial	1	0.31	0.9	1000	*traingdm*	0.7342	−0.3090	−1.1400
10	Factorial	1	0.01	0.6	700	*trainlm*	0.0311	−3.4700	−3.3500
11	Center	16	0.16	0.75	850	*trainlm*	0.0243	−3.7200	−3.8000
12	Factorial	1	0.31	0.9	1000	*trainlm*	0.0309	−3.4800	−3.3200
13	Center	16	0.16	0.75	850	*trainlm*	0.0256	−3.6700	−3.8000
14	Factorial	31	0.31	0.6	1000	*trainlm*	0.0251	−3.6800	−3.4300
15	Axial	31	0.16	0.75	850	*traingdm*	0.7759	−0.2537	−0.5696
16	Center	16	0.16	0.75	850	*traingdm*	0.2568	−1.3600	−1.8700
17	Factorial	31	0.01	0.9	700	*traingdm*	0.9892	−0.0109	−0.5575
18	Factorial	31	0.01	0.9	1000	*traingdm*	0.9729	−0.0275	−0.4826
19	Factorial	1	0.31	0.6	1000	*trainlm*	0.0312	−3.4700	−3.5000
20	Center	16	0.16	0.75	850	*traingdm*	0.2548	−1.3700	−1.8700
21	Factorial	31	0.31	0.9	1000	*trainlm*	0.0238	−3.7400	−3.5700
22	Factorial	31	0.31	0.9	1000	*traingdm*	0.9859	−0.0142	0.6025
23	Factorial	31	0.01	0.6	1000	*trainlm*	0.0250	−3.6900	−3.8000
24	Factorial	31	0.01	0.6	700	*trainlm*	0.0235	−3.7500	−3.5300
25	Factorial	31	0.31	0.6	700	*traingdm*	0.9744	−0.0259	−0.6975
26	Factorial	1	0.31	0.6	700	*traingdm*	0.0312	−3.4700	−2.9200
27	Factorial	1	0.01	0.9	700	*traingdm*	0.1039	−2.2600	−2.0500
28	Axial	16	0.01	0.75	850	*traingdm*	0.0721	−2.6300	−2.0500
29	Factorial	31	0.31	0.9	700	*traingdm*	0.9237	−0.0794	0.4133
30	Center	16	0.16	0.75	850	*trainlm*	0.0264	−3.6300	−3.8000
31	Factorial	1	0.31	0.9	700	*trainlm*	0.0309	−3.4800	−3.5700
32	Factorial	1	0.31	0.6	1000	*traingdm*	0.0313	−3.4600	−2.8300
33	Center	16	0.16	0.75	850	*traingdm*	0.1818	−1.7000	−1.8700
34	Axial	16	0.01	0.75	850	*trainlm*	0.0243	−3.7200	−3.5300
35	Axial	16	0.31	0.75	850	*traingdm*	0.8620	−0.1485	−1.1400
36	Factorial	1	0.01	0.9	1000	*trainlm*	0.0314	−3.4600	−3.1000
37	Axial	16	0.31	0.75	850	*trainlm*	0.0256	−3.6700	−3.5200
38	Center	16	0.16	0.75	850	*trainlm*	0.0248	−3.7000	−3.8000
39	Factorial	1	0.01	0.9	700	*trainlm*	0.0309	−3.4800	−3.2400
40	Axial	31	0.16	0.75	850	*trainlm*	0.0264	−3.6300	−3.5000
41	Center	16	0.16	0.75	850	*traingdm*	0.1030	−2.2700	−1.8700
42	Factorial	31	0.31	0.9	700	*trainlm*	0.0261	−3.6500	−3.6700
43	Factorial	1	0.01	0.9	1000	*traingdm*	0.1664	−1.7900	−1.8200
44	Axial	16	0.16	0.9	850	*traingdm*	0.1913	−1.6500	−1.4100
45	Factorial	1	0.01	0.6	700	*traingdm*	0.0615	−2.7900	−3.6600
46	Axial	16	0.16	0.75	700	*trainlm*	0.0262	−3.6400	−3.9000
47	Axial	1	0.16	0.75	850	*traingdm*	0.1048	−2.2600	−2.3500
48	Axial	16	0.16	0.75	1000	*traingdm*	0.0863	−2.4500	−1.9300
49	Factorial	31	0.01	0.9	1000	*trainlm*	0.0252	−3.6800	−3.7600
50	Axial	1	0.16	0.75	850	*trainlm*	0.0311	−3.4700	−3.2800
51	Factorial	31	0.01	0.6	700	*traingdm*	0.1022	−2.2800	−1.8500
52	Factorial	31	0.31	0.6	700	*trainlm*	0.0261	−3.6500	−3.2800
53	Factorial	1	0.31	0.9	700	*traingdm*	0.2292	−1.4700	−1.4800
54	Center	16	0.16	0.75	850	*traingdm*	0.1480	−1.9100	−1.8700
55	Factorial	1	0.01	0.6	1000	*traingdm*	0.0412	−3.1900	−3.6800
56	Factorial	31	0.01	0.9	700	*trainlm*	0.0243	−3.7200	−3.7400
57	Factorial	1	0.31	0.6	700	*trainlm*	0.0312	−3.4700	−3.5000
58	Center	16	0.16	0.75	850	*trainlm*	0.0259	−3.6500	−3.8000
59	Center	16	0.16	0.75	850	*traingdm*	0.0266	−3.6300	−1.8700
60	Axial	16	0.16	0.6	850	*traingdm*	0.0306	−3.4900	−2.9000

**Table 6 membranes-12-00726-t006:** The best network parameters for the FFNN-lm, FFNN-gdm, and FFNN-RSM models.

Method		Conventional			Proposed	
Model	FFNN-lm		FFNN-gdm		FFNN-RSM	
Network Parameters	Optimal Value	MSECV	Optimal Value	MSECV	Optimal Value	ln(MSECV)
A: Number of neurons	29	0.0238	8	0.0533	16	0.0264
B: Learning rate	0.7	0.0232	0.2	0.3403	0.16	0.0264
C: Momentum	0.5	0.0243	0.5	0.0297	0.75	0.0264
D: Number of epochs	300	0.0234	1000	0.0302	850	0.0264
E: Training function	*trainlm*		*traingdm*		*trainlm*	

**Table 7 membranes-12-00726-t007:** Overall performance accuracy of the FFNN-lm, FFNN-gdm, and FFNN-RSM models for the SMBR filtration system.

Method		Conventional			Proposed	
Model	FFNN-lm		FFNN-gdm		FFNN-RSM	
Parameters	MSE	R	MSE	R	MSE	R
Training	0.0223	0.9888	0.0296	0.9851	0.0237	0.9881
Testing	0.0253	0.9873	0.0303	0.9847	0.0265	0.9866
Training time (s)	151.02		392.68		233.22	
No. of runs or repetitions	60		60		60	
Total training time (s)	543.7				233.22	
Total no. of runs or repetitions	120				60	

## Data Availability

Not applicable.

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
