# Peer review of "Improved Artificial Neural Network Training Based on Response Surface Methodology for Membrane Flux Prediction"

_membranes, 2022, doi:10.3390/membranes12080726_

Round 1
Reviewer 1 Report
This work employed the improved artificial neural network training with response surface methodology optimization for membrane flux prediction. And the training with reduced training time and number of repetitions in achieving good model prediction for permeate flux of palm oil mill effluent. Several parameters, including the number of neurons, learning rate, momentum, epoch, and training functions, have been studied. This work is interesting and meaningful, providing a novel method to predict the membrane flux. The work is also well-organized and could be accepted in membranes.
Reviewer 2 Report
Review of membranes-1815972
This is an interesting multidiscipline manuscript about ANN (computer science) for the prediction of membrane flux (membrane science, part of chemical engineering), although based on two variables in the input layer, i.e. (1) transmembrane pressure and (2) permeate pump. However, there are a handful of issues to be clarified, as follows:
- Table 4 vs Equation 10: The significant variables from Table 4 are A, B, C, E, AE, BE. Why does Equation 10 displays A, B, C, D, E, AB, AC, AD, AE, BC, BD, BE, CD, CE, DE, AA, BB, CC, DD (and no EE)? Between Table 4 vs Equation 10, the variables must be those of significant values (p<0.05) so that the regression equation will be compact.
- Figure 6a: Please show the real number of MSECV for learning rate of 0.3 and 0.5, because those MSECV numbers are similar to the selected one (learning rate 0.7).
- Figure 7b: Please show the real number of MSECV for momentum of 0.1, 0.2, 0.3, 0.4, 0.6, and 0.7, because those MSECV numbers are similar to the selected one (momentum= 0.5).
- Figure 8c: Please show the real number of MSECV for number of epoch of 600, because those MSECV numbers are similar to the selected one (number of epoch= 1000).
- Please add list of abbreviations.
- Line 39: …is clogged by… --> not “is clogging”
- Line 40: …may vary… --> not “may varies”
- Line 96: These ANN topologies…--> not “these…topoLOGY”
- Line 96: …will affect… --> not “will affeCTS"
- Line 123: Please separate “100” and “kDa” with a space.
- Table 1, row 3: The unit for the range of PV-101 is Volt, with uppercase V, not lowercase v.
- Table 1, row 4: Define SLPM, what does it stand for?
- Table 1, row 8, 9, and 10: Write the unit in form of X Y-1, not X/Y, as L min-1.
- Table 1, row 12: kDa, with uppercase D (kilo Daltons).
- Line 134: Write the unit in form of X Y-1, not X/Y, as L m-2 h-1. --> check the one in line 147, this is the best way to write unit with more than one denominator.
- Line 181: Two thirds of…
- Line 183: One third of…
- Line 187: Please change “While,” with “On the other hand,”
- Line 243: K-fold --> with uppercase K, not lowercase k.
- Line 243: Please use American English only or British English only, but not both. Please change “optimisation” to be “optimization”, because the American style is dominant in this manuscript.
- Line 276: Change “is notify as” to be “is stated as”
- Line 310: Please change “Analysed” (British English) to be “Analyzed” (American English). In the next two lines (line 312), “generalization” (American English) is used.
- Line 375: Please change “utilised" to be “utilized”.
- Line 486: It may be concluded that…
- Line 530: Figure 12(a)-12(d).
- Line 591-592: Is there any supplementary information? If no, then delete these lines. If yes, then where is it?
- Reference: Delete the single quotation marks before and after the title of ALL article in the reference list in this manuscript.
- Reference 16: Modeling --> with uppercase M, because it is placed in the beginning of a sentence.
- Reference 16: De Sitter, K. --> referring to the original paper in this link https://doi.org/10.1016/j.seppur.2014.12.026 the surname of this author contains two words. So, please be careful.
- Reference 21: Delete “Elsevier BV”
- Reference 25 vs 26: These are IDENTICAL references. Delete reference 26! Please re-adjust the numbering sequence of reference 27 onwards!
- Reference 30: …fluidiZED.. --> ended with “ed”.
- Reference 31: Delete “Elsevier BV”
- Reference 32: TiO2 --> with subscripted 2
- Reference 33: Delete “Elsevier BV”
- Reference 34: Delete “Elsevier BV”
- Reference 35: Phormidium valderianum --> scientific names must be written in italic.
- Reference 38: What is the article number of this reference?
- Reference 39: Delete “Elsevier Ltd”
- Reference 39: What is the volume number of this reference?
- Reference 41: Strange title! Please correct accordingly --> Response surface optimization and artificial neural network modeling of microwave assisted natural dye extraction from pomegranate rind
- Reference 47: Delete “Elsevier Ltd.”
Round 2
Reviewer 2 Report
Review of membranes-1815972-v2
The authors have addressed all issues well. This manuscript can be accepted now.